# A G1-like state allows HIV-1 to bypass SAMHD1 restriction in macrophages

Petra Mlcochova[1], Katherine A Sutherland[1], Sarah A Watters[1], Cosetta Bertoli[2], Rob AM de Bruin[2], Jan Rehwinkel[3], Stuart J Neil[4], Gina M Lenzi[5], Baek Kim[5], Asim Khwaja[6], Matthew C Gage[7], Christiana Georgiou[7], Alexandra Chittka[7], Simon Yona[7], Mahdad Noursadeghi[1], Greg J Towers[1] & Ravindra K Gupta[1,8,*] [iD]

## Abstract

An unresolved question is how HIV-1 achieves efficient replication in terminally differentiated macrophages despite the restriction factor SAMHD1. We reveal inducible changes in expression of cell cycle-associated proteins including MCM2 and cyclins A, E, D1/D3 in macrophages, without evidence for DNA synthesis or mitosis. These changes are induced by activation of the Raf/MEK/ERK kinase cascade, culminating in upregulation of CDK1 with subsequent SAMHD1 T592 phosphorylation and deactivation of its antiviral activity. HIV infection is limited to these G1-like phase macrophages at the single-cell level. Depletion of SAMHD1 in macrophages decouples the association between infection and expression of cell cycle-associated proteins, with terminally differentiated macrophages becoming highly susceptible to HIV-1. We observe both embryo-derived and monocyte-derived tissue-resident macrophages in a G1-like phase at frequencies approaching 20%, suggesting how macrophages sustain HIV-1 replication *in vivo*. Finally, we reveal a SAMHD1-dependent antiretroviral activity of histone deacetylase inhibitors acting via p53 activation. These data provide a basis for host-directed therapeutic approaches aimed at limiting HIV-1 burden in macrophages that may contribute to curative interventions.

Keywords  cell cycle; histone deacetylase; HIV; macrophage; SAMHD1
Subject Categories  Microbiology, Virology & Host Pathogen Interaction
The EMBO Journal (2017) 36: 604–616

## Introduction

SAMHD1, a deoxynucleotide-triphosphate (dNTP) hydrolase, restricts HIV-1 reverse transcription (RT) through decreasing levels of dNTPs (Goldstone *et al*, 2011; Lahouassa *et al*, 2012; Schmidt *et al*, 2015). SAMHD1 phosphorylation at position T592 mediated by cyclin-dependent kinases CDK1/2 (Cribier *et al*, 2013; White *et al*, 2013) in actively dividing cells impairs the dNTP hydrolase activity and allows viral DNA synthesis to occur (Cribier *et al*, 2013; Arnold *et al*, 2015). Some lentiviruses have evolved countermeasures against SAMHD1; for example, the HIV-2/SIVsm lineage encodes the Vpx protein that degrades SAMHD1 and allows infection of otherwise SAMHD1-positive target cells (Kaushik *et al*, 2009; Hrecka *et al*, 2011; Laguette *et al*, 2011). How pandemic HIV-1 strains achieve efficient infection of terminally differentiated macrophages *in vivo* without a Vpx-like activity has remained a significant unresolved question that has limited our understanding of HIV tropism and pathogenesis (Watters *et al*, 2013).

Here, we reveal dynamic expression of cell cycle-associated proteins in non-dividing macrophages that do not culminate in DNA synthesis or mitosis. These changes are dependent on the canonical mitogen/growth factor-activated Raf/MEK/ERK signalling pathway and are sufficient to deactivate the potent HIV-1 restriction mediated by SAMHD1. Moreover, we show in two distinct populations of tissue-resident macrophages in mice that 20% of cells express a G1-like activation profile, providing not only an explanation for the ability of macrophages to sustain high levels of HIV-1 replication but also offering a therapeutic target for limiting HIV-1 burden in these vital innate immune cells.

1   Division of Infection and Immunity, University College London, London, UK
2   MRC Laboratory for Molecular Cell Biology, University College London, London, UK
3   Medical Research Council Human Immunology Unit, Radcliffe Department of Medicine, Medical Research Council Weatherall Institute of Molecular Medicine, University of Oxford, Oxford, UK
4   Division of Immunology, Infection and Inflammatory Disease, King's College, London, UK
5   Department of Pediatrics, Center for Drug Discovery, Emory School of Medicine, Atlanta, GA, USA
6   Research Department of Haematology, UCL, London, UK
7   Division of Medicine, University College London, London, UK
8   Africa Health Research Institute, KwaZulu Natal, South Africa
    *Corresponding author. Tel: +44 20 7679 2000; E-mail: ravindra.gupta@ucl.ac.uk

# Results

## Terminally differentiated macrophages stimulated to enter a G1-like phase are highly susceptible to HIV-1 infection

We observed that culture of human monocyte-derived macrophages (MDM) in foetal calf serum/FCS (stimulated cells), as opposed to human serum/HS (unstimulated cells), led to a significant increase in permissivity to HIV-1 infection (Figs 1A–C and EV1A–E). As expected, there was significant donor variation in absolute permissivity to HIV (Fig EV1B). An increase in viral permissivity under stimulating conditions was observed for single-round VSV-G-pseudotyped HIV-1 virus (Figs 1A and EV1A and B) and full-length infectious HIV-1 molecular clones (Figs 1B and C, and EV1C and D), including macrophage tropic viruses (BaL, YU-2), clinical HIV-1 isolates (Fig EV1C and D) and HIV-1 with capsid mutations known to alter interactions with cyclophilin and CPSF6 leading to altered reverse transcription, retargeted integration and triggering of innate sensing (Fig EV1E). The infection enhancement was observed post-entry at the step of reverse transcription (Figs 1B and EV1F). The effect of FCS on infection was lost when charcoal-stripped FCS was used but not when boiled FCS or a human serum/foetal calf serum mixture (1:1) was used, suggesting the existence of a heat stable stimulatory factor in FCS rather than an inhibitory factor in HS (Fig EV1A).

The differential effects of FCS and HS suggested an approach to uncovering mechanisms regulating HIV-1 permissivity in macrophages. We compared the transcriptomes of MDM differentiated in HS (unstimulated cells, UNSTIM) or FCS (stimulated cells, STIM) aiming to discover signalling pathways that contribute to higher HIV-1 permissivity. Comparison of transcriptional profiles for a predefined gene signature that discriminates macrophages from other cell types (Tomlinson *et al*, 2012) clearly shows that HS and FCS cultured MDM cluster together and are distinct from closely related myeloid cells (Fig 1D). Moreover, they express similar macrophage markers (Fig EV2). However, use of ingenuity pathway analysis to evaluate genes associated with high permissivity to HIV-1 infection revealed enrichment in a number of molecular/cellular functions including cell cycle regulation, growth and proliferation, and cell death or survival (Fig 1E). The top canonical pathway was enriched for genes encoding proteins involved in cell cycle regulation including cyclins (Figs 1F and EV3A, and Table EV1). These observations were validated at the protein expression level (Fig 1G). Stimulated cells showed an increase in D-type cyclins D1 and D3, which accumulate as cells progress through G1 phase (Baldin *et al*, 1993; Sherr, 1993, 1996). Cyclin D2 was below detectable levels under both conditions (Fig 1G). We also observed an increase in cyclins A and E, E2F6 and Geminin—all known to accumulate during cell cycle entry (Coverley *et al*, 2002; Bertoli *et al*, 2013; Fragkos *et al*, 2015). Of note, CDK1—a key player in cell cycle progression—was also upregulated in stimulated cells, along with MCM2 (minichromosome maintenance complex component 2) (Fig 1G and H), a replication origin licensing factor that is expressed from G1 (but not in G0) (Baldin *et al*, 1993; Sherr, 1996; Su & O'Farrell, 1997; Tsuruga *et al*, 1997; Musahl *et al*, 1998; Stoeber *et al*, 1998, 2001; Williams *et al*, 1998). Importantly, p27 expression was reduced following MDM stimulation (Fig 1G). p27 is a cyclin-dependent kinase inhibitor which is highly expressed in quiescent cells and which decreases after cell cycle re-entry (Sherr & Roberts, 1999).

We further compared stimulated and unstimulated cells at the single-cell level for expression of cell cycle-associated proteins, which are absent in G0/quiescent/terminally differentiated cells but (i) are found at all cell cycle phases (MCM2) (Masai *et al*, 2010), (ii) that accumulate in S and G2/M phases (Geminin) (Fragkos *et al*, 2015) and (iii) are specific to active DNA synthesis in cells and therefore a marker of S phase (EdU incorporation) (Fig 1H). Single cell-imaging analysis using an automated microscopic platform (Fig EV3B) showed a specific increase in each marker in stimulated versus unstimulated cells. The absolute number of cells positive for MCM2 was fivefold to 10-fold higher than cells positive for the S-G2-M marker Geminin (Fig 1H). Together with observed low levels of EdU incorporation over a 50-h period, these data suggest that stimulated MDM re-entered the cell cycle but the majority did not progress to S, G2/M phase (Figs 1H and EV3C). We confirmed further that stimulated MDM did not divide using classical propidium iodide (PI) and CFSE staining (Figs 1I and J, and EV3D). These data suggest that MDM are in a G0/quiescent/terminally differentiated state and that they are able to modulate expression of cell cycle-associated proteins, without measurable cell division or DNA synthesis. These early changes are consistent with transition to a G1-like phase and are associated with increased permissivity to HIV-1 infection in cultured MDM.

## Transition to a G1-like state is sufficient to regulate SAMHD1 antiviral activity in terminally differentiated macrophages

Given the known association between SAMHD1, a well-described HIV-1 restriction factor in myeloid cells and the cell cycle (Cribier *et al*, 2013; White *et al*, 2013; Pauls *et al*, 2014; Yan *et al*, 2015), we hypothesised that SAMHD1 antiviral activity might be regulated spontaneously in human MDM. To test this, we examined CDK1/2 expression knowing that CDK1/2 phosphorylates SAMHD1 to deactivate its capacity to restrict infection (Cribier *et al*, 2013; Welbourn *et al*, 2013; White *et al*, 2013). We found raised expression levels of CDK1 and pSAMHD1-T592 in stimulated MDM, but not CDK2, CDK4 or CDK6 (Fig 2A). dNTP levels were also increased by threefold to fourfold in stimulated MDM (Fig EV3E), also consistent with the reported activity of SAMHD1 (Lahouassa *et al*, 2012). Furthermore, exogenously induced degradation of SAMHD1 by co-infection with SIVmac virus-like particles bearing Vpx (VLP-vpx) (Fig 2B), or depletion of SAMHD1 by siRNA transfection (Fig 2C) led to a 10-fold increase in HIV-1 infectivity specifically in unstimulated MDM, with no change in infection in the permissive stimulated MDM where SAMHD1 is phosphorylated and thus already inactive against HIV-1.

To probe the reversibility of regulation of CDK1/2 and SAMHD1 in MDM, we changed serum 3 days before infection from unstimulating HS to stimulating FCS, and vice versa (Fig 2D and E). Non-stimulating conditions from day 3 onwards [unstim (day 3), Fig 2D and E] reduced MCM2 and CDK1 expressions (indicative of cells returning to a quiescent state), with decreased SAMHD1 phosphorylation and HIV-1 infection. Conversely, stimulating conditions from day 3 [stim (day 3), Fig 2D and E] or from day 7 (Fig EV3F) augmented HIV-1 infection and was associated with increased MCM2/CDK1 expression (indicative of cells re-entering the G1-like state) and SAMHD1 phosphorylation. No significant changes were

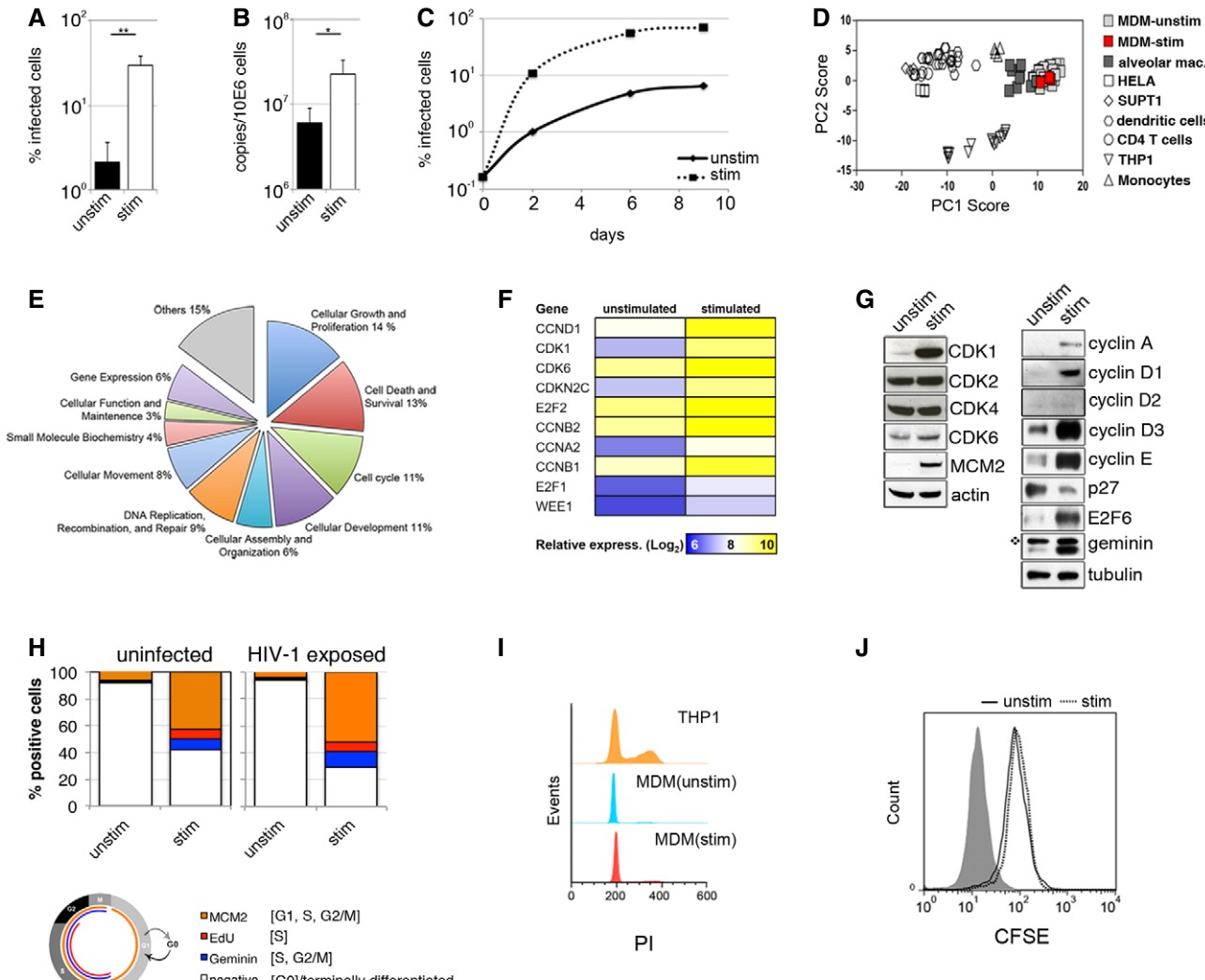

**Figure 1. Transitioning from G0- to a G1-like phase in terminally differentiated monocyte-derived macrophages (MDM).**

A   MDM were differentiated and cultured in RPMI complemented with MCSF and 10% human serum (unstim) or with MCSF and 10% foetal calf serum (stim). MDM were infected with VSV-G-pseudotyped HIV-1 expressing GFP, and the percentage of infected cells was quantified 48 h post-infection by FACS ($n = 3$, mean $\pm$ s.e.m.; **$P$-value $\leq 0.01$, unpaired $t$-test).

B   MDM were infected with HIV-1 BaL and DNA isolated 18 h post-infection for qPCR of late viral RT products ($n = 3$, mean $\pm$ s.e.m.; *$P$-value $\leq 0.05$, unpaired $t$-test).

C   Spreading infection in MDM. Cells were infected with HIV-1 BaL, stained for intracellular p24 and quantified by FACS.

D   Principal component analysis of expression data for macrophage-associated transcripts to compare relative clustering of stimulated and unstimulated MDM and a range of primary cells/cell lines.

E   Diagram of cellular and molecular functions associated with genes that show significant ($-\log_2(P\text{-value}) > 2$) transcriptional upregulation in stimulated MDM compared to unstimulated MDM.

F   Cell cycle-associated transcripts in stimulated and unstimulated MDM.

G   Immunoblot of cell cycle-associated proteins expressed in MDM. Star indicates non-specific band. This Western blot quantification is from Donor 1 in Fig 2A. The same blots were used in Fig 2A to allow comparison of different cell cycle-associated proteins, as well as SAMHD1.

H   Uninfected MDM or MDM exposed for 48 h to VSV-G HIV-1 GFP (HIV-1 exposed) were stained for cell cycle-associated proteins MCM2, Geminin and EdU incorporation (EdU added to MDM 50 h prior to analysis). On average, $10^4$ cells in each experiment were recorded and analysed using Hermes WiScan cell-imaging system and ImageJ. Diagram of the cell cycle pathway with associated markers is also shown.

I   Cell cycle analysis by quantitation of DNA content by flow cytometry. Cycling THP-1 cells and both unstimulated and stimulated MDM were labelled by propidium iodide (PI) and analysed by FACS.

J   CFSE loaded MDM were cultured for 4 days to determine cell division/proliferation by FACS.

Source data are available online for this figure.

---

observed for CDK2 expression or phosphorylation (Fig 2E). These data suggest that this process is reversible in terminally differentiated macrophages.

To further explore the regulation of SAMHD1 in MDM, we mapped the signalling pathway responsible for SAMHD1 phosphorylation by using well-characterised specific inhibitors of kinases

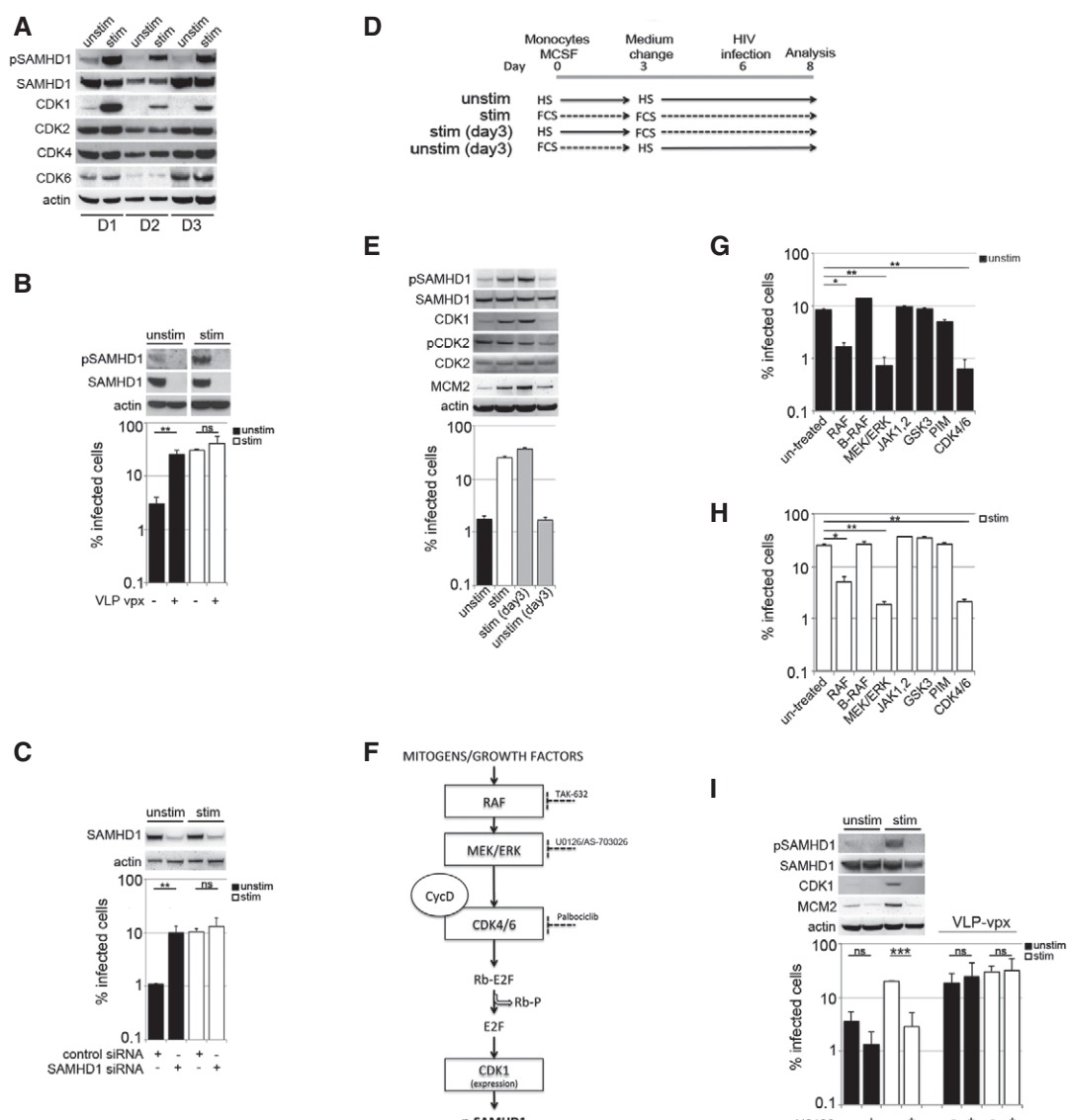

**Figure 2. Bidirectional transitions shape SAMHD1-mediated restriction of HIV-1.**

A MDM from three donors (D1, D2, D3) were used for immunoblotting to detect SAMHD1 and CDK proteins. The blot for D1 is the same as that in Fig 1G in order to facilitate comparison of different cell cycle-associated proteins and SAMHD1.

B MDM were co-infected with VSV-G HIV-1 GFP and SIVmac virus-like particles containing vpx (VLP-vpx). Cells from a representative donor were used for immunoblotting. The percentage of infected cells was quantified by FACS 48 h post-infection ($n = 2$, mean $\pm$ s.e.m.; **$P$-value $\leq 0.01$; (ns) non-significant, unpaired $t$-test).

C MDM were transfected with control or pool of SAMHD1 siRNAs and infected 3 days later with VSV-G-pseudotyped HIV-1 GFP. Cells from a representative donor were used for immunoblotting. The percentage of infected cells was quantified by FACS 48 h post-infection ($n = 2$, mean $\pm$ s.e.m.; **$P$-value $\leq 0.01$; (ns) non-significant, unpaired $t$-test).

D Experimental approach used to model MDM bidirectional G0–G1-like transitions. MDM were as follows: (unstim) cultured in HS or (stim) cultured in FCS as described in Materials and Methods; [stim (day 3)] grown in HS conditions for 3 days and changed to stimulating FCS conditions for 3 days; [unstim (3 days)] grown in stimulating FCS condition for the 3 days and changed to non-stimulating HS for the remaining 3 days.

E Single round of infection of MDM with full-length HIV-1 BaL. Cells were used for immunoblotting to detect CDKs, SAMHD1 and MCM2 proteins. Graph is a representative example of $n \geq 3$, mean $\pm$ s.e.m.

F Proposed signalling pathway leading to SAMHD1 phosphorylation in stimulated MDM.

G, H Unstimulated (G) and stimulated (H) MDM were treated with inhibitors of RAF (2 $\mu$M), B-RAF (3 $\mu$M), MEK1/2 (AS-703026, 1 $\mu$M), JAK 1–3 (1 $\mu$M), GSK3 (2 $\mu$M), PIM 1–3 (3 $\mu$M) and CDK4/6 (1 $\mu$M) for 18 h before infection and infected with VSV-G HIV-1 GFP. The percentage of infected cells was quantified by FACS 48 h post-infection. Graphs are representative example of $n \geq 3$, mean $\pm$ s.e.m., *$P$-value $\leq 0.05$; **$P$-value $\leq 0.01$; calculated from triplicates, unpaired $t$-test).

I MDM were treated with a MEK/ERK inhibitor (U0126, 10 $\mu$M) for 18 h before infection and where indicated VLP-vpx was added at the time of infection. Percentage of infected cells were detected by FACS 48 h post-infection. Cells were used for immunoblotting to detect CDKs, SAMHD1 and MCM2 proteins ($n = 3$, mean $\pm$ s.e.m.; (ns) non-significant; ***$P$-value $\leq 0.001$, unpaired $t$-test).

Source data are available online for this figure.

(Fig 2F–I). We identified the involvement of the Raf/MEK/ERK kinase cascade in the regulation of HIV restriction (Figs 2F–I and EV3G). Of note, the B-Raf inhibitor PLX4032 (active only against the V600E B-Raf mutant observed in cancer cells) was used as a negative control to demonstrate specificity of Raf inhibition (Fig 2G and H). To pharmacologically block the putative signal activated by FCS, we also used a highly specific inhibitor of MEK/ERK, U0126, reasoning that this would lead to suppression of HIV-1 infection in a SAMHD1-dependent manner (Fig 2F). Indeed, MEK/ERK inhibition substantially inhibited HIV-1 infection of MDM, and loss of HIV-1 permissivity correlated with SAMHD1 dephosphorylation, CDK1 and MCM2 downregulation (Fig 2I). Critically, SAMHD1 depletion by VLP-vpx (Fig 2I) completely rescued HIV-1 infection from the inhibitory effect of U0126. This was further confirmed by inhibiting the signalling pathway downstream from MEK/ERK using a specific inhibitor of CDK4/6 (Fig EV3G). These data illustrate, for the first time, that the Raf/MEK/ERK signalling pathway regulates SAMHD1 antiviral activity in terminally differentiated macrophages.

### G1-like phase macrophages are preferential targets for HIV at the single-cell level

We next employed a high-throughput single-cell co-localisation analysis, aiming to measure the association between SAMHD1, markers of cell cycle progression and HIV-1 infection at the single-cell level (Fig 3). We infected MDM with VSV-G-pseudotyped HIV-1 and 48 h post-infection stained the cells for MCM2 or for EdU incorporation (added at the time of infection) to monitor active DNA synthesis and analysed cells by using an automated microscopic system (Fig 3A–E). As expected, we found that MCM2 expression correlated with HIV infection under both stimulatory and non-stimulatory conditions (Fig 3A–C and G–J). This analysis illustrated that macrophages in a G1-like state, measured by MCM2 expression, are preferential targets for HIV at the single-cell level. Stimulation by FCS simply increased the number of MCM2-positive cells and therefore increased the number of permissive target cells for HIV-1 infection (Fig 3B, C, E, G, and H). This explains the increased permissivity of stimulated MDM cultures in Fig 1A. Importantly, even though stimulatory conditions also increased the number of cells positive for EdU incorporation (Fig EV4A), the population of cells actively synthesising DNA was a minority (< 7%) and this increase was not statistically significant. As shown earlier, MDM very rarely divide, even under stimulatory conditions (Figs 1I and J, and EV3D). As HIV infection did not co-localise with EdU incorporation in single-cell analyses, we conclude that entry into S phase and DNA synthesis is not required for enhanced HIV-1 infection (Figs 3C and J, and EV4B), consistent with a G1-like permissivity window.

Critically, degradation of SAMHD1, through co-infection with SIVmac VLP-vpx, or SAMHD1 depletion using RNAi, completely abrogated the association between infection of MDM and MCM2 expression (Figs 3D, E and G–J, and EV4C). Indeed, log odds ratios comparing frequencies of HIV-1 infection in MCM2-positive and MCM2-negative cells were high, indicating highly specific infection of MCM2-positive cells, but not MCM2-negative cells. These log odds ratios were significantly reduced after SAMHD1 depletion indicating that SAMHD1 was responsible for the poor permissivity of the MCM2-negative cells (Fig 3J). There was no such evidence for

preferential permissivity of Edu-positive cells (Fig 3J). Importantly, Semliki Forest virus (SFV), an alpha virus with an RNA genome which replication is dNTP-independent, did not preferentially infect MDM expressing MCM2 (Figs 3F and J, and EV4D and E). As predicted, infection of MDM by lentiviruses naturally encoding Vpx genes (HIV-2, SIVsmE543) also showed significant reduction in log odds ratios (Fig 3J).

### Tissue-resident macrophages commonly reside in a G1-like phase and are preferential HIV-1 targets

Human MDM are a widely used model for primary macrophages, but their generation relies on *in vitro* differentiation from monocytes. We therefore investigated whether tissue-resident macrophages could be observed in the G1-like state we describe and whether this was associated with increased HIV-1 permissivity. SAMHD1 is conserved in mice and has anti-HIV-1 activity (Behrendt *et al*, 2013; Zhang *et al*, 2014) that, like human SAMHD1, is regulated by phosphorylation (Wittmann *et al*, 2015). We first isolated mouse microglia (tissue-resident, yolk sac-derived macrophages of the brain) and observed between 5 and 20% of these cells expressing MCM2 immediately following isolation (Fig EV5). However, MCM2 staining in microglia appeared to diminish rapidly following isolation from brain tissue, precluding analysis of the correlation between MCM2 and HIV-1 infection. We therefore isolated tissue-resident peritoneal macrophages (of bone marrow origin) (Fig 4) from both wild-type (WT) and SAMHD1 knock-out (KO) mice (Rehwinkel *et al*, 2013). We infected these cells with VSV-G-pseudotyped HIV-1 and at 48 h post-infection inspected the cells for specific macrophage markers F4/80 and CD11b (Fig 4A and B), as well as expression of MCM2 and EdU incorporation (Fig 4C–I). We observed that 20% of macrophages from both SAMHD1 WT and KO mice were positive for MCM2 (Fig 4C and H), but ≤ 1% of cells incorporated EdU. As before, this suggested a G1-like state that does not progress into S phase similar to the situation in human MDM (Fig 4D and I). Critically, and as before, we observed preferential infection of MCM2-positive cells from WT but not SAMHD1 negative cells (Fig 4E–G, J and K). There was no correlation between infection and EdU in macrophages from either WT or KO mice (Fig 4K).

### HDAC inhibitors induce a SAMHD1-dependent block to HIV-1 in human MDM

Histone deacetylase inhibitors (HDACi) can reactivate latent HIV in a variety of experimental systems (Wightman *et al*, 2012). HDACi also induce cell cycle arrest and induce a differentiated phenotype and/or apoptosis in most carcinoma cell lines (Marks *et al*, 2000; Komatsu *et al*, 2006; West & Johnstone, 2014), but the effect of these drugs on HIV infection in primary human macrophages is not known. To test this, we treated MDM with increasing concentrations of HDACi SAHA (vorinostat) or panobinostat, infected the cells with VSV-G-pseudotyped HIV-1 and measured the percentage of infected cells 48 h later. HDACi substantially inhibited HIV-1 infection of both stimulated and unstimulated MDM (Figs 5A and EV6A) with no observed cytotoxicity (Fig 5B). For all further experiments, we used only stimulated MDM (Fig 5B–F), where the majority of cells are in a G1-like phase and expressing MCM2 and inactive phosphorylated SAMHD1. Here, the loss of HIV-1 permissivity following

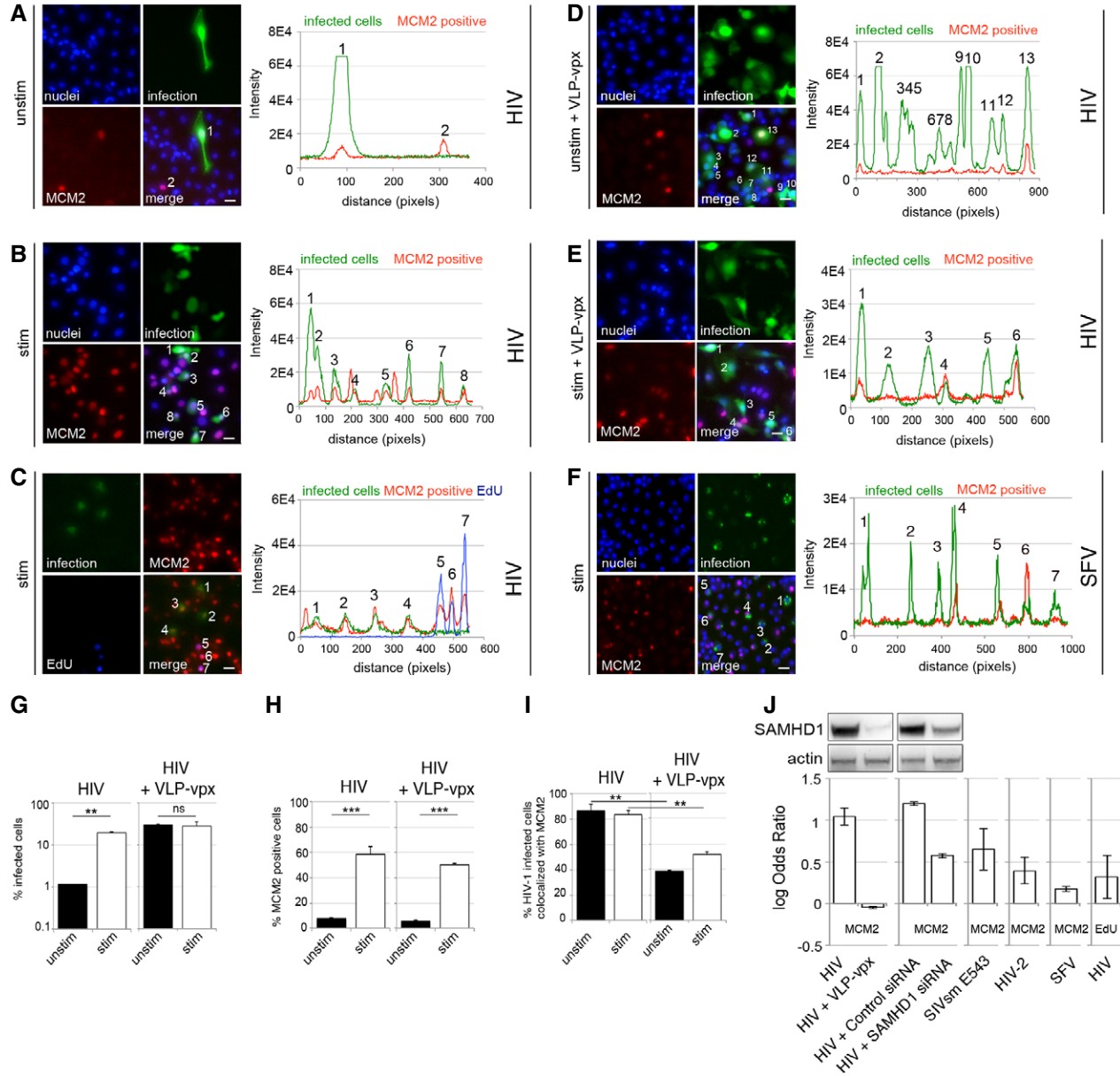

**Figure 3. HIV-1 tropism for G1-like MDM at the cellular level.**

A–C  MDM were infected with VSV-G-pseudotyped HIV-1 GFP, stained and recorded and analysed for infection, MCM2 expression and co-localisation between infection and MCM2 protein and EdU incorporation 48 h post-infection using Hermes WiScan cell-imaging system. Random microscopic fields and plot profiles show an example of immunofluorescence intensity along infected cells. Scale bars: 20 μm.

D, E  MDM were infected with VSV-G-pseudotyped HIV-1 GFP in the presence of SIVmac virus-like particles containing vpx (VLP-vpx), stained and recorded 48 h post-infection using Hermes WiScan cell-imaging system. Random microscopic fields and plot profile show an example of immunofluorescence intensity along infected cells. Scale bars: 20 μm.

F  MDM were infected with Semliki Forest virus (SFV). Random microscopic fields and plot profile show an example of immunofluorescence intensity along infected cells. Scale bar: 20 μm.

G–I  MDM were infected with VSV-G-pseudotyped HIV-1 GFP in the presence or absence of VLP-vpx, stained and recorded 48 h post-infection. On average, $10^4$ cells in each experiment were recorded and analysed for infection (G), MCM2 expression (H) and co-localisation between infection and MCM2 protein (I) using Hermes WiScan cell-imaging system and ImageJ ($n = 3$, mean ± s.e.m.; (ns) non-significant; **$P$-value ≤ 0.01, ***$P$-value ≤ 0.001, unpaired $t$-test).

J  Log odds ratios calculated from quantifications of stimulated MDM show association of infection with MCM2/EdU (1, high association; 0, no association) ($n ≥ 3$, mean ± s.e.m.). Immunoblot shows expressions levels of SAMHD1 in these experiments. EdU was added to cells at the time of infection.

HDACi treatment correlated with loss of MCM2 (Fig 5C and D), CDK1 and SAMHD1 dephosphorylation, suggesting transition of MDM from G1-phase to a non-permissive, inactive state (Fig 5D).

HDACi treatment of MDM was also associated with increased expression of cell cycle regulators p27 and p53 (Figs 5D and EV6B) (Sherr & Roberts, 1999; Wang *et al*, 2015). SAMHD1 depletion using

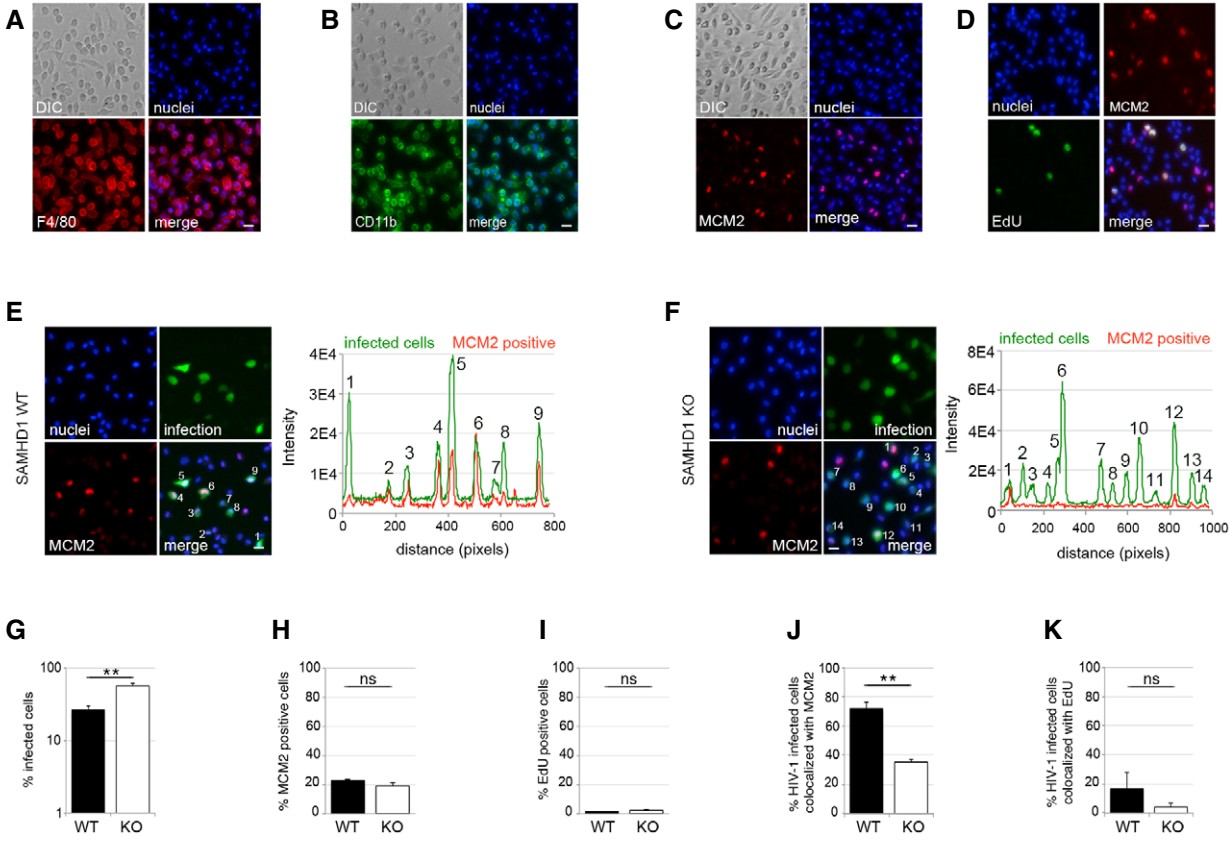

**Figure 4. HIV-1 tropism for murine tissue-resident macrophages at the cellular level.**

A–D   Peritoneal macrophages from 6- to 10-week-old C57BL/6 mice were stained for the macrophage markers (A) F4/80, (B) CD11b, (C) MCM2 or (D) active DNA synthesis (EdU) 2 h after isolation. Scale bars: 20 μm.

E–K   Peritoneal macrophages were isolated from 6- to 10-week-old WT or SAMHD1$^{-/-}$ (KO) mice. Cells were infected with VSV-G-pseudotyped HIV-1 GFP for 48 h and analysed for (G) infection, (H) MCM2 expression, (I) EdU incorporation (EdU was added at the time of infection), and co-localisation between infection and (J) MCM2 protein or (K) active DNA synthesis (by EdU incorporation). (E, F) Random microscopic fields and plot profiles show an example of immunofluorescence intensity along infected cells. Scale bars: 10 μm. On average, 10$^4$ cells in each experiment were recorded and analysed using Hermes WiScan cell-imaging system and ImageJ ($n \geq 2$, mean ± s.e.m.; (ns) non-significant; **$P$-value $\leq$ 0.01, unpaired $t$-test).

siRNA (Fig 5E and F) or by VLP-vpx (Fig EV6C–F) completely rescued HIV-1 infection from the inhibitory effect of HDACi. These data indicate that HDACi possess antiretroviral activity that is SAMHD1-dependent in macrophages.

## Discussion

HIV infects terminally differentiated tissue macrophages within the gut, lung, lymph nodes and central nervous system (CNS) (Gonzalez-Scarano & Martin-Garcia, 2005; Yukl *et al*, 2014; Cribbs *et al*, 2015). Macrophages represent the dominant cellular target in the CNS, and compartmentalised viral populations have been documented in CSF, consistent with T-cell-independent replication in this "sanctuary site" (Schnell *et al*, 2011). As HIV-infected macrophages are able to release virus over extended periods, they are not only a potential source of drug-resistant virus, but also a likely barrier to achieving remission/cure (Watters *et al*, 2013).

Here, we propose that SAMHD1 is the dominant host factor controlling post-entry permissivity to infection of non-dividing MDM.

We demonstrate that the SAMHD1 phosphorylation status at T592 controlling antiviral activity is naturally dynamic in primary human MDM, providing a window of opportunity for HIV-1 infection without a requirement for an anti-SAMHD1 countermeasure. Intriguingly, the dynamic phosphorylation of SAMHD1 by CDK1 is associated with expression of proteins typically associated with cell cycle control. These include cyclins D1 and D3, cyclins A and E, E2F6, MCM2 and Geminin. Therefore, despite the fact that that the vast majority MDM in our experiments do not actually synthesise DNA (lack of EdU incorporation) or divide (by PI and CFSE assays), we find that MDM can be stimulated to exit G0 phase and enter a G1-like state. This transition with associated CDK1 upregulation and T592 phosphorylation of SAMHD1 renders macrophages permissive to HIV-1.

It is not clear why macrophages should transition between a classical G0 state and a G1-like state. One possibility is that nucleotide accumulation is required for repair of damaged DNA, as has been observed in terminally differentiated post-mitotic neurons (Kruman *et al*, 2004).

Our finding that microglia and peritoneal macrophages reside in the G1-like state may explain how macrophages can sustain HIV-1

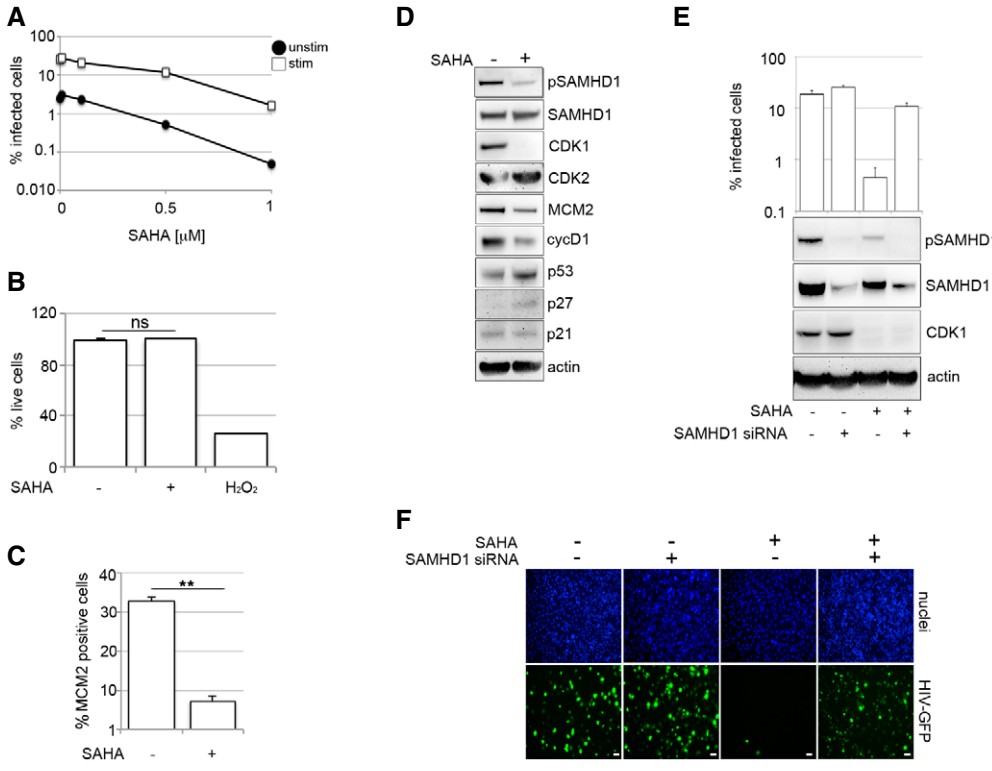

**Figure 5. HDAC inhibitors induce a SAMHD1-dependent block to HIV-1 infection in macrophages.**

A    MDM were treated with increasing concentrations of the HDAC inhibitor SAHA 18 h before infection with VSV-G-pseudotyped HIV-1 GFP. The percentage of infected cells was measured using FACS 48 h post-infection. Graph is representative example of $n \geq 3$, mean $\pm$ s.e.m.

B    Stimulated MDM were treated with 1 $\mu$M SAHA and 48 h later stained for viability using a cell viability assay (LIVE/DEAD) ($n \geq 2$, mean $\pm$ s.e.m.; (ns) non-significant, unpaired $t$-test). $H_2O_2$ was used as a positive control to induce cell death.

C    Stimulated MDM were treated with 1 $\mu$M SAHA, and MDM were recorded and analysed for MCM2 protein expression using Hermes WiScan ($n = 3$, mean $\pm$ s.e.m.; **$P$-value $\leq$ 0.01, unpaired $t$-test).

D    Stimulated MDM were treated with 1 $\mu$M SAHA, MDM were lysed, and immunoblotting was performed to detect cell cycle-associated proteins.

E, F    MDM were transfected with control or pool of SAMHD1 siRNAs and infected 3 days later with VSV-G-pseudotyped HIV-1 GFP. Cells from a representative donor were (E) lysed and used for immunoblotting or (F) inspected for infection. Scale bars: 20 $\mu$m. On average, $10^4$ cells in each experiment were recorded and analysed using Hermes WiScan cell-imaging system and ImageJ ($n \geq 2$, mean $\pm$ s.e.m.).

Source data are available online for this figure.

infection in the absence of T cells within a humanised mouse model (Honeycutt *et al*, 2016). Importantly, we could recapitulate the G1-like HIV-1 permissivity window in freshly isolated peritoneal macrophages from mice. This demonstrates that the dynamic SAMHD1 regulation is conserved in mice and that *in vivo* macrophages are likely to experience the transitions we describe. We found that cells taken directly from mouse brain, in contrast to peritoneum, frequently reverted from G1-like phase to the quiescent state. This suggests either that microglia are intrinsically programmed to more readily revert back to a quiescent state than peritoneal macrophages or that maintenance of the G1-like state in microglia requires specific local tissue factors.

The brain is sometimes referred to as a "sanctuary" site where isolated primate lentiviral replication occurs in myeloid cells, including perivascular macrophages and microglia (Williams *et al*, 2001; Schnell *et al*, 2009, 2010; Micci *et al*, 2014). Our biological insights into dynamic changes essential for HIV-1 infection in macrophages may be particularly relevant for HIV neuropathology given observations that symptomatic patients on long-term fully suppressive

combination antiretroviral therapy (ART) can have isolated viral replication in the CNS (Schnell *et al*, 2009, 2010; Canestri *et al*, 2010; Hammond *et al*, 2016). We speculate that the brain microenvironment could modulate G0- to G1-like transitions in brain myeloid cells and thereby impact HIV replication in the CNS and HIV persistence in this compartment.

In the light of the above, induction of a state of cellular resistance to HIV infection by host-directed therapy in addition to ART is highly desirable as we search to achieve sustained HIV remission. This may be particularly important if combination ART cannot prevent ongoing low-level replication and evolution (Lorenzo-Redondo *et al*, 2016). Here, we reveal a hitherto unrecognised SAMHD1-dependent antiretroviral activity of HDACi in macrophages. In addition to their known effects on gene transcription (West & Johnstone, 2014) and HIV reactivation from latency through histone modification, HDACi activate SAMHD1 antiviral activity via p53-induced cell cycle exit, CDK1 depletion and loss of SAMHD1 phosphorylation. The ability to reactivate latent virus and to prevent infection of new target cells is a major advantage of these

agents as they enter clinical trials as candidates for achieving functional HIV cure.

We mapped activation of the Raf/MEK/ERK signalling cascade after addition of FCS. However, we have not identified the factor(s) leading to transition to the G1-like phase and differential effects of FCS and HS on SAMHD1 activity in MDM. FCS and HS differ in a number of ways, including concentration of growth factors, hormones and endotoxin. Charcoal stripping but not boiling of FCS abrogated the enhanced permissivity arising from FCS culture, consistent with a hormone or fatty acid being the putative active agent. Furthermore, it is currently unclear whether the differential effect on SAMHD1 is due to differences between foetal and adult sera or differences between human and bovine sera. Indeed, we could not reproduce the effect of FCS using human cord blood-derived serum (data not shown).

Recently, Badia *et al* reported that MDM cultured in the presence of GM-CSF induced the expression of cyclin D2 with downstream changes in SAMHD1 phosphorylation and increased susceptibility to HIV-1 infection (Badia *et al*, 2016). In particular, they were unable to detect cyclin D2 expression by Western blot in either HS or FCS cultured M-CSF stimulated macrophages, despite differences in susceptibility to HIV-1 and SAMHD1 phosphorylation. Concordantly, we report undetectable cyclin D2 under both culture conditions, consistent with a cyclin D2-independent mechanism activated by FCS that is driven by the Raf/MEK/ERK signalling pathway and CDK1 regulation.

In summary, we have answered the question of how HIV-1 infects SAMHD1 expressing macrophages without encoding a Vpx-like SAMHD1 antagonist. We have found that MDM and tissue-resident macrophages exist in their typical terminally differentiated/quiescent/G0 state, but also in a G1-like state. Those in G0 can be stimulated into re-entering the G1-phase of the cell cycle, without evidence for cell division, accompanied by deactivation of SAMHD1 by phosphorylation. This allows efficient HIV-1 replication without the need to degrade SAMHD1. SAMHD1 shutdown provides a window of opportunity for HIV-1 infection, and we believe that this is how HIV-1 can infect macrophages *in vivo*. The translational potential of this new knowledge is exemplified by our demonstration that HDAC inhibitors can block HIV infection at the pre-integration stage through activation of SAMHD1, thereby protecting these critical immune cells.

# Materials and Methods

### Reagents, inhibitors, antibodies, plasmids

Tissue culture media and supplements were obtained from Invitrogen (Paisley, UK), and tissue culture plastic was purchased from TPP (Trasadingen, Switzerland). FCS (FBS) was purchased from Biosera (Boussens, France) and Sigma (Sigma, St. Louis, MO, USA). Charcoal-stripped FCS was purchased from Sigma. Human serum from human male AB plasma was of USA origin and sterile-filtered (Sigma). All chemicals were purchased from Sigma unless indicated otherwise. Kinase inhibitors used: CDK4/6 inhibitor (PD 0332991, Palbociclib) from Sigma; MEK/ERK inhibitor U0126 from Calbiochem (San Diego, USA); JAK 1–3 (ruxolitinib), PIM 1–3 (AZD1897), GSK3 (CT99021), and MEK1/2 (AS-703026), RAF (TAK-632),

B-RAF (PXL4032) from Selleckchem (Houston, TX, USA), SAHA (vorinostat) from Sigma and panobinostat from Cayman Chemicals (Ann Arbor, MI, USA). Antibodies used were as follows: anti-cdc2 (Cell Signaling Technology, Beverly, MA, USA); anti-CDK2 (H-298, Santa Cruz Biotechnology); anti-pCDK2(Thr160) (Bioss Inc., MA, USA); anti-CDK4 (DCS156, Cell Signaling Technology); anti-CDK6 (B-10, Santa Cruz Biotechnology); anti-SAMHD1 (ab67820, Abcam, UK), beta-actin (ab6276, abcam, UK); mouse anti-MCM2 (BM-28, BD Biosciences, UK); and rabbit anti-MCM2 (SP85) from Sigma; pSAMHD1 (a kind gift from M. Benkirane) and ProSci (Poway, CA, USA); anti-Geminin (NCL-L-Geminin, Leica); anti-mouse F4/80 and CD11b (kind gift from S. Yona, UCL); anti-human CD68, CD14, CD163, CD80, CD86, CD40 (kind gift from M. Noursadeghi). All infectious molecular clones were obtained from the NIH AIDS Research and Reference Reagent Program (Germantown, MD, USA).

### Cell lines and viruses

TZM-bl HeLa and 293T cells were cultured in DMEM complete (DMEM supplemented with 100 U/ml penicillin, 0.1 mg/ml streptomycin and 10% FCS). HIV-1 full-length virus stocks were generated by DNA plasmid transfection of 293T using Fugene HD (Promega UK Ltd, UK) according to the manufacture's protocol. Viral supernatants were harvest 48 h post-transfection and filtered through 0.45-μm pore-size filters and stored at −80°C. Clarified viral supernatants were analysed by p24 ELISA (AIDS and Cancer Virus Program NCI-Frederick, MD, USA) for HIV-1 p24 antigen quantification. SIVmac virus-like particles containing Vpx were prepared as previously described (Goujon *et al*, 2008). VSV-G HIV-1 GFP virus was produced by transfection of 293T with GFP-encoding genome CSGW, packaging plasmid p8.91 and pMDG as previously described (Besnier *et al*, 2002). SFV was kind gift from M. Mazzon (UCL). VSV-G HIV-2 GFP virus was produced as previously described (Ylinen *et al*, 2005), and VSV-G SIVsmE543 GFP virus was kind gift from G. Towers (UCL).

### Monocyte isolation and differentiation

PBMC were prepared from HIV-seronegative donors (after informed consent was obtained), by density-gradient centrifugation (Lymphoprep, Axis-Shield, UK). MDM were prepared by adherence with washing of non-adherent cells after 2 h, with subsequent maintenance of adherent cells in RPMI 1640 medium supplemented with 10% human serum or 10% foetal calf serum and MCSF (10 ng/ml) for 3 days and then differentiated for a further 4 days in RPMI 1640 medium supplemented with 10% human/foetal calf sera without M-CSF. Human AB serum (Sigma) was used to prepare unstimulated cells or FCS (Biosera or Sigma) to prepare stimulated cells.

### Infection of primary cells using full-length and VSV-G-pseudotyped HIV-1 viruses

MDM were infected with 50 ng of p24 of each virus for 4 h. Cells were washed in PBS, and new complete medium was added. MDM were fixed in ice-cold acetone–methanol (1:1 [vol/vol]) 2 days post-infection, and infected cells identified by staining for p24 protein using a 1:1 mixture of the anti-p24 monoclonal antibodies EVA365

and EVA366 (NIBSC, Center for AIDS Reagents, UK) and a secondary goat anti-mouse beta-galactosidase-conjugated antibody (SouthernBiotech, AL, USA) and visualised by X-Gal (5-bromo-4-chloro-3-indolyl-β-d-galactopyranoside) staining (Promega). Virus-infected cells were detected by light microscopy. Alternatively, MDM were fixed in 3% PFA, permeabilised by saponin and stained for intracellular p24 using anti-p24 FITC-conjugated antibody (Santa Cruz Biotechnology, USA). The percentage of infected cells was monitored by flow cytometry using BD FACSCalibur (BD Biosciences, UK) and analysed by CellQuest (BD Biosciences) and FlowJo software (Tree Star, OR, USA). GFP containing VSV-G-pseudotyped HIV-1 was added to cells (MDM and tissue-resident macrophages from mice), and after 4-h incubation, removed and cells were washed in culture medium. The percentage of infected cells was determined 48 h post-infection by flow cytometry using BD FACSCalibur (BD Biosciences, UK) and analysed by CellQuest (BD Biosciences) and FlowJo software (Tree Star, OR, USA) or by Hermes WiScan-automated cell-imaging system (IDEA Bio-Medical Ltd. Rehovot, Israel) and analysed using MetaMorph and ImageJ software.

### Quantitative PCR for total HIV DNA quantitation

Total HIV DNA was detected as previously described (Mlcochova *et al*, 2014).

### Measurement of HIV-1 entry (BlaM-Vpr assay)

HIV-1 entry was measured as previously described (Mlcochova *et al*, 2014).

### TZM-bl assay

Supernatants from BaL-infected MDM were harvested, filtered using 0.45-μm pore-sized filters and used to infect TZM-bl cells. Luciferase activity of the TZM-bl cells was measured 24 h post-infection using the Steady Glo Firefly Luciferase assay (Promega) and GloMax96 Luminometer (Promega).

### SDS–PAGE and immunoblots

Cells were lysed in reducing Laemmli SDS sample buffer containing PhosSTOP (Phosphatase Inhibitor Cocktail Tablets, Roche, Switzerland) at 96°C for 10 min and the proteins separated on NuPAGE® Novex® 4–12% Bis–Tris Gels. Subsequently, the proteins were transferred onto PVDF membranes (Millipore, Billerica, MA, USA), the membranes were quenched, and proteins detected using specific antibodies. Labelled protein bands were detected using Amersham ECL Prime Western Blotting Detection Reagent (GE Healthcare, USA) and Amersham Hyperfilm or AlphaInnotech CCD camera. Protein band intensities were recorded and quantified using AlphaInnotech CCD camera and AlphaView software (Protein-Simple, San Jose, CA, USA).

### RNA microarrays

Total RNA was purified from cell lysates collected in RLT buffer (Qiagen) using the RNeasy Mini kit (Qiagen). Samples were processed for Agilent microarrays, and data were normalised as previously described (Chain *et al*, 2010). Microarray data are available in the ArrayExpress database under accession number E-MTAB-2985 for stimulated MDM (differentiated in FCS) and E-TABM-1206 for all other cell types presented in this study. The macrophage-associated gene expression module was used as previously described (Tomlinson *et al*, 2012), and genes encoding nuclear proteins were derived from the Gene Ontology consortium database (AmiGO v1.8, http://amigo1.geneontology.org/cgi-bin/amigo/go.cgi), under accession number GO:0005634, filtered for cellular compartment and for *Homo sapiens*, to give 6,164 unique genes with gene symbol annotations. Ingenuity pathway analysis (Ingenuity® Systems, www.ingenuity.com) was used to identify significantly enriched pathways among 170 genes with > twofold expression in stimulated and unstimulated MDM.

### Cell proliferation assay, propidium iodine and CFSE labelling

Cell proliferation was measured by tracking new DNA synthesis using Click-iT® EdU Alexa Fluor® 488 Kit (Invitrogen). EdU was added to culture medium at 5 μM for 2 h or 2–4 days depending on experiment. Labelled cells were detected using the Hermes WiScan-automated cell-imaging system (IDEA Bio-Medical Ltd. Rehovot, Israel) and analysed using MetaMorph and ImageJ software. For propidium iodine (PI) staining: cells were fixed in 70% ethanol, treated with RNAse and stained with PI (0.1 mg/ml), monitored by flow cytometry using BD FACSCalibur and analysed by CellQuest and FlowJo software. 5 μM CFSE was added to MDM according to CellTrace™ CFSE Cell Proliferation Kit manufacturer protocol (ThermoFisher, Waltham, MA, USA), and cells were left in culture to show potential cell division for an additional 6 days. CFSE labelling was conducted according to CellTrace™ CFSE Cell Proliferation Kit manufacturer protocol. Cells were monitored by flow cytometry using BD FACSCalibur and analysed by CellQuest and FlowJo software.

### Immunofluorescence

MDMs were fixed in 3% PFA, quenched with 50 mM NH₄Cl and permeabilised with 0.1% Triton X-100 in PBS or 90% methanol. After blocking in PBS/1% FCS, MDMs were labelled for 1 h with primary antibodies diluted in PBS/1% FCS, washed and labelled again with Alexa Fluor secondary antibodies for 1 h. Cells were washed in PBS/1% FCS and stained with DAPI in PBS for 20 min. Labelled cells were detected using Hermes WiScan-automated cell-imaging system (IDEA Bio-Medical Ltd. Rehovot, Israel) and analysed using MetaMorph and ImageJ software.

### SAMHD1 knock-down by siRNA

$1 \times 10^5$ MDM differentiated in MCSF for 4 days were transfected with 20 pmol of siRNA (L-013950-01, Dharmacon) using Lipofectamine RNAiMAX Transfection Reagent (Invitrogen). Transfection medium was replaced after 18 h with RPMI 1640 medium supplemented with 10% human AB serum or 10% FCS and cells cultured for additional 3 days before infection.

### Tissue-resident brain macrophages

All work conformed to United Kingdom Home Office legislation (Scientific Procedures Act 1986) (https://www.gov.uk/government/publications/consolidated-version-of-aspa-1986). Adult CD1 mice aged 8–12 weeks were anaesthetised using 20% Pentoject® and perfused with ice-cold PBS. Microglia were isolated as previously described (Denk *et al*, 2016). Briefly, brains were dissected and homogenised using dounce tissue homogeniser in 0.2% BSA supplemented Hank's balanced salt solution (HBSS). Microglia were isolated using a Percoll density gradient (37% versus 70%). Cells were subsequently counted using haemocytometer and plated in 96-well plates at $2.5 \times 10^4$–$5 \times 10^4$ cells/well. Cells were maintained in DMEM/F12 with 10% FCS at 37°C, 5% $CO_2$ until further use.

### Tissue-resident peritoneal macrophages

Six- to ten-week-old C57BL/6 wild-type and knock-out mice were used in accordance with the UK Home Office Scientific Procedures Act 1986. Peritoneal cavities were washed with PBS supplemented with 3 mM EDTA. Following gentle massage, the cavity was opened by abdominal incision and lavage fluid collected. Fluid was centrifuged 500 *g* for 5 min, and cells were re-suspended in DMEM + 10% FCS and cultured for 2 h before staining or infection.

### dNTP measurement

The dNTP levels in the relevant cell types were measured by the HIV-1 RT-based dNTP assay as previously described (Diamond *et al*, 2004).

### Ethics statement

Adult subjects provided written informed consent. Primary Macrophage & Dendritic Cell Cultures from Healthy Volunteer Blood Donors has been reviewed and granted ethical permission by the National Research Ethics Service through The Joint UCL/UCLH Committees on the Ethics of Human Research (Committee Alpha) 2 December 2009; reference number 06/Q0502/92.

**Expanded View** for this article is available online.

### Acknowledgements

This work was funded by a Wellcome Trust fellowship to RKG (WT108082AIA) and the National Institute for Health Research University College London Hospitals Biomedical Research Centre. GJT is funded by Wellcome Trust Senior Biomedical Research Fellowship 108183, the European Research Council under the European Union's Seventh Framework Programme (FP7/2007–2013)/ERC grant agreement number 339223 and the Medical Research Council. This work was also partially supported by USA National Institutes of Health grants, AI049781 (B.K.) and GM104198 (B.K.). We would also like to thank Mark Wainberg, Richard Goldstein, Anne Bridgeman, Jennifer Roe, Laura Hilditch, Deenan Pillay, Arne Akbar, Rob Sellar, Daniel Hochhauser and Clare Jolly for helpful advice and reagents.

### Author contributions

PM, RKG, SY, GJT, MN, AC, AK, MCG, BK, SJN, AC, RAMB, JR designed experiments; RKG, PM, MN, GJT wrote the manuscript; PM, SAW, MCG, SY, GML, CG, KAS, CB performed experiments; and PM, RKG, SY, GJT, MN, AC, AK, MCG, BK, SJN, AC, CB, KAS, JR, RAMB analysed data.

### Conflict of interest

The authors declare that they have no conflict of interest.

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
