## [Review Process File · The EMBO Journal]

Manuscript EMBO-2016-96025

A G1-like state allows HIV-1 to bypass SAMHD1 restriction in macrophages

Petra Mlcochova, Katherine A Sutherland, Sarah A Watters, Cosetta Bertoli, Robertus AM de Bruin, Jan Rehwinkel, Stuart J Neil, Gina M Lenzi, Baek Kim, Asim Khwaja, Matthew C Gage, Christiana Georgiou, Alexandra Chittka, Simon Yona, Mahdad Noursadeghi, Greg J Towers, Ravindra K Gupta

Corresponding author: Ravindra Gupta, UCLL

Review timeline:

Submission date:	08 November 2016
Editorial Decision:	05 December 2016
Revision received:	12 December 2016
Accepted:	21 December 2016

Editor: Karin Dumsrei

Transaction Report:

1st Editorial Decision

05 December 2016

Thanks for submitting your manuscript to The EMBO Journal. Your study has now been seen by three good experts in the field and their comments are provided below.

As you can see from the comments, the referees find the analysis interesting and suitable for publication here. They raise a few relative minor concerns that shouldn't involve too much additional work to sort out.

Let me know if we need to discuss anything further.

REFeree REPORTS

Referee #1:

The authors noted that HIV-1 transduction efficiency was greater in macrophages cultured in fetal calf serum than in human serum. Differential expression profiling (FCS vs HS) revealed differences in cell cycle regulators that were confirmed by western blots for cyclins A, D, E, CDK1, MCM2, p27, geminin, and E2F6, though cells remained blocked before entry into S phase. The cell cycle effects were associated with SAMHD1 phosphorylation, and, via a number of convincing experiments, the cell cycle-dependent, Raf/MEK/ERK-dependent, SAMHD1 phosphorylation was shown to explain the effects of the FCS on HIV infectivity. Individual macrophages were assessed by microscopy and it was shown that transduction correlated with cell cycle markers for G1;

correlation was lost with Vpx or other controls. Experimental results were replicated using macrophages from WT and SAMHD1 KO mice. The mouse experiments also confirmed that G1 macrophages are not a tissue culture artifact. HDAC inhibitors blocked the effect of FCS.

This is a very important manuscript that clearly explains how HIV-1 is capable of infecting macrophages, despite the fact that the virus lacks a protein, such as Vpx, that counteracts the antiviral effect of SAMHD1. We have only trivial suggestions to improve the manuscript.

1. Page 3, line 2: The statement that myeloid and resting T cells express SAMHD1 is kind of irrelevant, since it implies that other cell types do not express it. The second sentence about SAMHD1 phosphorylation in dividing cells then is kind of confusing.
2. Page 4, line 14: The relevance to the experiments here of the HIV-1 capsid mutants is not clear. It seems an irrelevant distraction.
3. Page 4, line 16: With respect to the charcoal stripping, did the authors ever try to mix the two sera together to see which phenotype is dominant? Is the sera heat-inactivated? Does heat kill the activity?
4. Page 4, line 22: Why call them stimulated and unstimulated? It would be clearer to the reader if they were simply called FCS vs HS.

Referee #2:

Mlcochova and colleagues examined why HIV-1 is capable of infecting macrophages although these express SAMHD1 and the virus does not express an antagonist of this restriction factor. They show that parts of the macrophages are in a G1-like phase that is associated with expression of various cell cycle associated proteins including CDK1 that phosphorylates and hence inactivates SAMHD1. The authors also provide evidence that these events are induced by FCS treatment via stimulation of the Raf/MEK/ERK cascade. Finally, the authors provide evidence that HDAC inhibitors prevent HIV-1 infection of macrophages in a SAMHD1 dependent manner.

The experiments are for most part well performed and controlled. One limitation of the study is that the novelty of the findings seems somewhat limited. It is known that macrophages become susceptible to HIV-1 infection after inactivation of the restriction factor SAMHD1 by CDK1. Furthermore, Badia and colleagues have recently shown that cell cycle proteins affect SAMHD1-mediated HIV-1 restriction in macrophages. This paper is mentioned in the discussion but not properly cited. Thus, the major novel aspects here seem to be the observation that FCS activates CDK1 via the Raf/MEK/ERK pathway and the inhibitory effect of HDAC inhibitors. Thus, should be clarified. Furthermore, some flaws in the manuscript need to be fixed.

Specific points

1. Abstract: the authors state "SAMHD1 activation by phosphorylation". However, as correctly stated elsewhere in the manuscript phosphorylation of SAMHD1 impairs its anti-HIV activity. To avoid confusion statement should be checked for accuracy throughout (see e.g. pg. 11, line 17 "inactive unphosphorylated SAMHD1").
2. The authors generally determined percentages of infected cells. For most part that's fine but the should examine whether induction of the G1-like phase is also associated with higher levels of infectious virus production by the infected cultures.
3. The infection rates in unstimulated macrophages are much higher in the experiment shown in Figure 2G than in 1A. This discrepancy should be discussed.
4. Did none of the inhibitors used to generate the data shown in Figure 2 show cytotoxic effects?
5. The inducing factor remains elusive and it seems surprising that it is found in FCS but not in human sera. The authors mention that they "could not reproduce the effect of FCS using human cord

blood derived serum". Further detail should be provided and they should discuss possible reasons for the differential effects of human and bovine sera (preparation, species-specific differences,...?). Obviously, identification of the factor would clearly increase the significance of the study.

6. At several places commas and spaces are missing.

Referee #3:

This is a beautiful and focused study addressing a long standing question in the field. They identified a population of non-dividing monocytes derived macrophages and resident macrophages that expresses markers of G1 as opposed to G0 macrophages. Transition from G0 to G1 overcomes the SAMHD1 restriction activity by inducing its phosphorylation at T592. The experiments are smartly designed, consistent, and fully support the conclusions. The study will certainly represents a forward step towards our understanding of macrophages as target for HIV-1 and their role in HIV-1 pathogenesis.

This reviewer is highly supportive for publication in The EMBO Journal.

Straightforward study. The experiments are well designed and appropriate. They fully support the conclusions.

I therefore have no additional experiments to ask for.

1st Revision - authors' response

12 December 2016

Referee #1:

The authors noted that HIV-1 transduction efficiency was greater in macrophages cultured in fetal calf serum than in human serum. Differential expression profiling (FCS vs HS) revealed differences in cell cycle regulators that were confirmed by western blots for cyclins A, D, E, CDK1, MCM2, p27, geminin, and E2F6, though cells remained blocked before entry into S phase. The cell cycle effects were associated with SAMHD1 phosphorylation, and, via a number of convincing experiments, the cell cycle-dependent, Raf/MEK/ERK-dependent, SAMHD1 phosphorylation was shown to explain the effects of the FCS on HIV infectivity. Individual macrophages were assessed by microscopy and it was shown that transduction correlated with cell cycle markers for G1; correlation was lost with Vpx or other controls. Experimental results were replicated using macrophages from WT and SAMHD1 KO mice. The mouse experiments also confirmed that G1 macrophages are not a tissue culture artifact. HDAC inhibitors blocked the effect of FCS.

This is a very important manuscript that clearly explains how HIV-1 is capable of infecting macrophages, despite the fact that the virus lacks a protein, such as Vpx, that counteracts the antiviral effect of SAMHD1. We have only trivial suggestions to improve the manuscript.

1. Page 3, line 2: The statement that myeloid and resting T cells express SAMHD1 is kind of irrelevant, since it implies that other cell types do not express it. The second sentence about SAMHD1 phosphorylation in dividing cells then is kind of confusing.

Response: We agree with this point and thank the reviewer for pointing it out. We have now removed the sentences in question on page 3.

2. Page 4, line 14: The relevance to the experiments here of the HIV-1 capsid mutants is not clear. It seems an irrelevant distraction.

Response: we know that cofactor interactions impact RT and are important for efficient HIV infection in macrophages. We therefore wished to test the hypothesis that the effect of FCS might be due to regulation of host co-factors sensitive to mutations in capsid. Even though this did not prove to be the case we believe that these are important negative data and would like to keep them in manuscript as long as the editor agrees.

3. Page 4, line 16: With respect to the charcoal stripping, did the authors ever try to mix the two sera together to see which phenotype is dominant? Is the sera heat-inactivated? Does heat kill the activity?

Response: We did mix the sera and found that the FCS phenotype is dominant, see figure below which we have now included as Extended view Figure 1A and added relevant text (page 4 lines 86-89).

The sera used were all heat inactivated and boiling FCS failed to abrogate the phenotype. We have added a figure to extended view as well as text in the results section to reflect this addition (page 4 lines 86-89).

Figure: Monocyte derived macrophages (MDM) were differentiated and cultured in RPMI complemented with MCSF and 10% Human Serum (UNSTIM) and changed at day 3 for 10% Fetal Calf Serum (STIM), 10% charcoal stripped FCS (CS), 10% FCS boiled for 5min (FCS boil), or 10% of 1:1 mixed HS:FCS (unstim:stim), HS:FCS boil (unstim:FCS boil) and infected with VSV-G pseudotyped HIV-1 expressing GFP. Percentage of infected cells was quantified 48h post-infection by FACS.

4. Page 4, line 22: Why call them stimulated and unstimulated? It would be clearer to the reader if they were simply called FCS vs HS.

Response: We debated this point for some time but arrived at ‘stimulated versus unstimulated’ because FCS stimulates the canonical Raf/MEK/ERK pathway to induce cell cycle entry, thereby making the figures easier to understand for the reader. As the other 2 reviewers did not comment we would prefer to keep as is unless the editor decides otherwise.

Referee #2:

Mlcochova and colleagues examined why HIV-1 is capable of infecting macrophages although these express SAMHD1 and the virus does not express an antagonist of this restriction factor. They show that parts of the macrophages are in a G1-like phase that is associated with expression of various cell cycle associated proteins including CDK1 that phosphorylates and hence inactivates SAMHD1. The authors also provide evidence that these events are induced by FCS treatment via stimulation of the Raf/MEK/ERK cascade. Finally, the authors provide evidence that HDAC inhibitors prevent HIV-1 infection of macrophages in a SAMHD1 dependent manner.

The experiments are for most part well performed and controlled. One limitation of the study is that the novelty of the findings seems somewhat limited. It is known that macrophages become susceptible to HIV-1 infection after inactivation of the restriction factor SAMHD1 by CDK1. Furthermore, Badia and colleagues have recently shown that cell cycle proteins affect SAMHD1-mediated HIV-1 restriction in macrophages. This paper is mentioned in the discussion but not properly cited. Thus, the major novel aspects here seem to be the observation that FCS activates CDK1 via the Raf/MEK/ERK pathway and the inhibitory effect of HDAC inhibitors. Thus, should be clarified.

Response: We are pleased that the reviewer feels 2 major aspects of the paper are novel. However, we disagree about novelty related to permissivity of macrophages and cell cycle associated proteins. We show that early cell cycle entry (from G0 to G1) is associated with dramatic changes in SAMHD1 dependent permissivity to HIV at the single cell level. This central finding in our paper was realised by rigorous quantification of co-staining in tens of thousands of cells, with a number of controls. The focus of the Badia et. al. paper was on the inhibitory effect of GMCSF on permissivity of macrophages acting via CDK1 and cyclin D2. We apologise for not citing the paper correctly in the bibliography, which we have now rectified (line 324). Thus our data are highly novel, explaining

how HIV replicates without the need for a SAMHD1 antagonist in both monocyte derived and tissue resident macrophages.

Furthermore, some flaws in the manuscript need to be fixed. Specific points

1. Abstract: the authors state "SAMHD1 activation by phosphorylation". However, as correctly stated elsewhere in the manuscript phosphorylation of SAMHD1 impairs its anti-HIV activity. To avoid confusion statement should be checked for accuracy throughout (see e.g. pg. 11, line 17 "inactive unphosphorylated SAMHD1").

Response: We thank the reviewer for pointing this mistake out – we have now corrected it and checked the paper.

2. The authors generally determined percentages of infected cells. For most part that's fine but the should examine whether induction of the G1-like phase is also associated with higher levels of infectious virus production by the infected cultures.

Response: we have measured infectious virus production in infected cultures and we have now presented these data as Extended view Figure 1D.

3. The infection rates in unstimulated macrophages are much higher in the experiment shown in Figure 2G than in 1A. This discrepancy should be discussed.

Response: the discrepancy is due to donor variation in absolute infection rate and the fact that we have used a representative donor for Fig 2G. The donor variability among 12 donors is demonstrated in Extended view figure 1B. In this figure one can find multiple donors with similar susceptibility as in Fig 2G. We have added text explaining that there is donor variability (lines 79-80).

4. Did none of the inhibitors used to generate the data shown in Figure 2 show cytotoxic effects?

Response: We thank the reviewer for this question. All inhibitors were carefully titrated on MDM and non-toxic, effective concentrations of compounds were determined and used for further experiments. The lack of cytotoxic effect is evidenced by the fact that SAMHD1 degradation rescued viral titre completely, as documented in the Figure below.

Figure: Stimulated MDM were treated with inhibitors of RAF (2mM); MEK1/2 (AS-703026, 1mM); and CDK4/6 (1mM) 18h before infection. MDM were coinfecting with VSV-G HIV-1 GFP and SIVmac Virus Like Particles containing vpx (VLP-vpx, degrades SAMHD1). Percentage of infected cells were quantified by FACS 48h post-infection.

5. The inducing factor remains elusive and it seems surprising that it is found in FCS but not in human sera. The authors mention that they "could not reproduce the effect of FCS using human cord blood derived serum". Further detail should be provided and they should discuss possible reasons for the differential effects of human and bovine sera (preparation, species-specific differences,...?). Obviously, identification of the factor would clearly increase the significance of the study.

Response: We agree that identification of the factor is important and we are undertaking an extensive study, but we would like to emphasise that the FCS and HS were primarily used as a tool to study macrophages in G1 and G0. Data from 19 human cord blood derived serum samples are shown below. No statistical significance was detected. We have also now discussed possible differences between human and foetal calf serum in the text (lines 314-320).

Figure: MDM were cultured in human serum, FCS or human cord blood derived serum (individual donors in capital letters) for 3 days and infected with VSV-pseudotyped HIV-1GFP. Percentage of infected cells was determined 48h postinfection by FACS.

6. At several places commas and spaces are missing.

Response: we have now proof read the paper again paying particular attention to this.

Corresponding Author Name: Ravindra Gupta

Journal Submitted to: EMBO J

Manuscript Number: EMBOJ-2016-96025